# Effects of High-Intensity Interval Training on Sleep: A Systematic Review and Meta-Analysis

**DOI:** 10.3390/ijerph182010973

**Published:** 2021-10-19

**Authors:** Leizi Min, Dizhi Wang, Yanwei You, Yingyao Fu, Xindong Ma

**Affiliations:** Division of Sport Science & Physical Education, Tsinghua University, Beijing 100084, China; mlz20@mails.tsinghua.edu.cn (L.M.); wangdz19@mails.tsinghua.edu.cn (D.W.); youyw20@mails.tsinghua.edu.cn (Y.Y.); fyy19@mails.tsinghua.edu.cn (Y.F.)

**Keywords:** high-intensity interval training, sleep, meta-analysis, systematic review

## Abstract

Objectives: To use a quantitative approach to examine the effects of high-intensity interval training (HIIT) interventions on sleep for adults. Methods: PubMed, Ebsco, Web of Science, Cochrane Central Register of Controlled Trials, the China National Knowledge Infrastructure, and Wanfang Data were searched from their inception to December 2020. Intervention studies with a control group that examined the effects of HIIT interventions on sleep were included in this meta-analysis. The risk of bias was assessed using the tool provided by the Cochrane Handbook for Systematic Reviews of Interventions. Effect sizes (ESs), calculated as weighted mean difference (WMD) and standardized mean difference (SMD), were used to examine the effects of objective outcomes and subjective outcomes separately. Results: A large increase in sleep quality (SQ) reflected by the Pittsburgh Sleep Quality Index global scores [WMD = −0.90, 95%CI (−1.72, −0.07), *p* = 0.03, *n* = 8] and a small-to-medium favorable effect on sleep efficiency (SE) [SMD = 0.43, 95%CI (0.20, 0.65), *p* = 0.0002, *n* = 10] were found after HIIT intervention. In addition, sub-analyses results suggest that ESs were moderated by the type, duration and frequency, as well as the length of the HIIT intervention. Conclusions: HIIT may be a promising way to improve overall subjective SQ and objective SE. PROSPERO, protocol registration number: CRD42021241734.

## 1. Introduction

Sleep is essential for maintaining the health and well-being of individuals of all ages. For children and adolescents, the functions of sleep include promoting growth, learning, and cognitive development [1]. For middle-aged and older adults, poor sleep quality is linked to poor living quality, low work efficiency, and many diseases, such as obesity, anxiety and depression [2,3,4]. However, an epidemiological survey shows that, with the growth of age, there are growing numbers of people experiencing sleep disorders for various reasons, which is the main cause of poor sleep quality [5,6]. The appropriate sleep duration for adults is 7–9 h per night, but the sleep duration of modern people is difficult to reach the recommended sleep duration, and more than one-third of people sleep less than 6 h per night [7]. With the outbreak of COVID-19, a researcher found that lack of sleep affected not only disease severity but also prognosis [8]. Thus, it is of paramount importance to promote sleep quality for people of all ages.

A new study has investigated the detrimental associations of poor sleep with all-cause and cause-specific mortality risks exacerbated by low physical activity, suggesting likely synergistic effects [9]. While exercise training in improving sleep quality has received accelerated attention, the best type, frequency, intensity and duration of exercise still need to be explored. 

As lack of time is often a key barrier to regular exercise participation [10], HIIT provides a time-metabolic effective alternative, especially for adults who are busy with their work. Ross et al. found that after 24 weeks of intensive exercise, abdominal obesity decreased by about 4.6 cm (range −6.2–3.0 cm), while after the same time of aerobic exercise, abdominal obesity decreased by about 3.6 cm (range −5.1–2.2 cm). Meanwhile, in multiple randomized controlled trials, including on the skeletal muscles, vascular system, respiration, autonomic function, cardiac function, exercise ability, inflammation, quality of life, VO_2_peak, HIIT has better improvement than MCT [11]. In addition, the reduced time required to achieve health improvements may be one important aspect for many obese adults, showing potential for higher rates of adherence [12,13].There is not only evidence that HIIT has physiological effects, but also evidence that HIIT plays a psychological role, such as improving anxiety and depression [14]. In fact, the positive feedback to complete a task increases enjoyment and, in this way, adherence to exercise [15]. To the best of our knowledge, although previous qualitative systematic reviews have observed exercise, especially aerobic exercise to have beneficial effects on sleep, no quantitative meta-analysis based on experimental studies has been conducted about the effects of HIIT on sleep. 

Therefore, the primary purpose of this study was to evaluate the effects of HIIT on sleep quality (sleep latency, sleep efficiency, total sleep time, etc.). Secondary objectives were to summarize the effects of HIIT on depression, anxiety, and health-related quality of life. 

## 2. Materials and Methods

### 2.1. Protocol and Registration

This systematic review with meta-analysis was registered in the International Prospective Register of Systematic Reviews (PROSPERO) trial registry (CRD42021241734). Additionally, the general guidelines of the Preferred Reporting Items for Systematic Reviews and Meta-Analysis (PRISMA) Statement were followed [13].

### 2.2. Literature Search Methods

PubMed, Ebsco, Web of Science, Cochrane Central Register of Controlled Trials, the China National Knowledge Infrastructure, and Wanfang Data were searched from their inception to December 2020. The search strategy combined the 2 groups of relevant terms: (1) “HIIT”, “high intensity interval training”, “high intensity interval exercise”, “high-intensity intermittent training” and “high-intensity training”; (2) “sleep” and “sleep quality”. 

### 2.3. Inclusion and Exclusion Criteria

Studies were included if they met all of the following criteria without language restrictions: (1) Participants: adults (aged 18–75 years old), including athletes, healthy non-athletes, cancer patients, obese patients and sedentary people; (2) Intervention: “high intensity interval training (HIIT)”, “high intensity interval exercise”, “high-intensity intermittent training” or “high-intensity training”; (3) Outcomes: the primary outcome measures were sleep quality, including subjective sleep quality assessment scale (Pittsburgh Sleep Quality Index, Epworth Sleepiness Score, Insomnia Severity Index and Apnoea-hypopnea Index) or objective sleep measurement tools (polysomnography, Actiwatch and Sense Wear Armband); the secondary outcome measures were anxiety, depression and health-related quality of life. (4) Study design: randomized controlled trials (RCT) and crossover studies.

Studies were excluded if they met one of the following criteria: (1) Participants were adolescents under 18 years of age and elderly people over 75; (2) Outcomes were not available; (3) Study design were intervention studies without an appropriate control group (e.g., rest control or usual care control) allowing for the discernment of causal effects.

### 2.4. Assessment of Quality

Two independent reviewers (L.M. and D.W.) assessed the quality of included studies using the risk of bias tool provided by the Cochrane Handbook for Systematic Reviews of Interventions [16]. A total of 7 items were assessed for each study: selection bias (random sequence generation and allocation concealment), performance bias (blinding of participants and personnel) detection bias (blinding of outcome assessment), attrition bias (incomplete outcome data), reporting bias (selective reporting) and other bias. For each item, there are three potential bias judgments: “low risk” “high risk” or “unclear risk”. Disagreements between the reviewers were resolved by discussion, with the involvement of a third author (Y.Y.) where necessary. 

### 2.5. Data Collection Process

Data were extracted on a prespecified worksheet. This included study characteristics (source, first author’s name, and year of publication), participant characteristics (age, gender and number), details of interventions (type, frequency, intensity, duration, length, and control group), mean of primary and secondary outcomes with standard deviation including pre- and post-exercise outcomes. For studies that reported partial data or only presented data by graphs, we sent emails to authors for initial data.

### 2.6. Statistical Analysis

We performed statistical analyses by the Cochrane Review Manager software (version 5; https://training.cochrane.org/online-learning/core-software-cochrane-reviews/revman, accessed on 9 January 2021). The chi-square test and I^2^ statistic were used to determine whether there was heterogeneity among the results. The fixed-effect model was used when *p* ≥ 0.1 and I^2^ < 50%, which indicates that multiple similar studies are homogeneous. When *p* ≤ 0.1 and I^2^ ≥ 50%, the random-effects model was used and subgroup analysis (such as different length of intervention, frequency and duration of intervention) was performed to look for sources of heterogeneity [17]. For continuous data, the weighted mean difference (WMD) was used as a statistic for effect analysis if the data were obtained from the same measurement tool. Standardized mean difference (SMD) [16] was used as the statistic for effect analysis if different measurement tools were used for the same variable. All effect sizes (ESs) were expressed as 95% confidence interval (95% CI) and interpreted as small (0.2), medium (0.5), or large (0.8). The significance level was set at *p* < 0.05. Lastly, funnel plots were used to evaluate publication bias if there were more than 10 included studies. Trim and Fill [18] analyses were conducted to recalculate estimated ES and confidence intervals based on the newly symmetrical distribution, and *z*-tests were applied to evaluate significance.

## 3. Results

### 3.1. Study Selection

The initial search yielded 678 studies (148 from PUBMED, 131 from Cochrane, 54 from Ebsco, 326 from WOS, 3 from WANFANG and 16 from CNKI). After removing 421 duplicates, we conducted a title and abstract search in the remaining 257 articles, which resulted in 41 studies. As for 216 excluded studies, the main causes were no data on the outcome of interest, no appropriate control group or participants < 18 years. After evaluating the abstracts of each study, we excluded 17 studies for failing the priori inclusion/exclusion criteria. Three articles were excluded after a close reading of the text for unavailable data. The final analysis consisted of 21 articles and 755 participants (Figure 1). 

### 3.2. Study Characteristics

Characteristics of the selected studies are presented in Table 1. Of them, three studies were conducted in Australia [19,20,21], three studies were conducted in Norway [22,23,24], two studies were conducted in the United States [25,26], two studies were conducted in Italy [27,28], two studies were conducted in Canada [29,30], two studies were conducted in Belgium [31,32], and one study was conducted in China [33], Germany [34], Switzerland [35], France [36], Spain [37], Chile [38], and Cyprus [39], respectively. In addition, 20 studies were written in English and one study was published in Chinese. The sample sizes ranged from 10 to 134; the intervention duration ranged from a single bout to 20 weeks; the HIIT sessions ranged from 8 min to 35 min; and the HIIT frequencies ranged from one time/week to seven times/week. Among them, 20 studies implemented HIIT-only interventions and one study implemented HIIT combined with resistance training interventions. According to the characteristics of included articles, we divided the type of HIIT intervention into cycling and other because the forms of HIIT are varied. In addition, we defined HIIT intervention duration ≥16 min as high, <16 min as low; HIIT frequency >3 times per week as high, ≤3 times per week as low; and HIIT intervention length >8 weeks as high, ≤8 weeks as low. Moreover, according to the source of participants, we divided them into the general population and patients. 

### 3.3. Quality of the Evidence

Figure 2 and Figure 3 show that the risk bias of 21 studies was moderate. Ten studies described the methods of randomization, four reported details on allocation concealment, six studies used blinding of participants and personnel, four studies used blinding of outcome measures, and two studies were at high risk of attrition bias. No study was at high risk of reporting bias and no evidence shows other bias in these studies. Finally, the results demonstrated that the quality of the evidence was acceptable.

### 3.4. Primary Outcomes

PSQI refers to the Pittsburgh Sleep Quality Index, which is a classic subjective measurement of sleep quality. A total score, as well as seven subscale scores (daytime sleepiness, sleep disturbance, sleep duration, sleep efficiency, sleep latency, sleep medication use, and subjective sleep quality) were provided by PSQI [42]. The sleep quality increases with the decrease of PSQI global score. A total of eight articles reported this outcome after removing one primary study from the pooled analysis because of bias [33]. Figure 4 shows a forest plot of each individual study. These articles [21,26,29,30,31,33,38,40] included 383 participants while only three of them provided subscale scores. Overall, HIIT has large beneficial effects on sleep quality reflected by PSQI global scores [WMD = −0.90, 95%CI (−1.72, −0.07), *p* = 0.03, *n* = 8]. Since I^2^ > 50, a random-effects model was used to calculate them. A funnel plot of PSQI was shown in Figure 5a, which appears to be asymmetrical. Moreover, the Insomnia Severity Index (ISI) is a commonly used brief self-report tool measuring insomnia [43]. A higher score suggests more severe insomnia and worse sleep status. The effects were highly statistically significant in the category of ISI [WMD = −1.65, 95%CI (−2.43, −0.87), *p* < 0.0001, *n* = 3]. However, there were not enough available studies [23].

As for objective outcomes, sleep efficiency (SE), sleep latency (SL), total sleep time (TST), and time in bed (TB) were assessed by different measurements such as Sense Wear Armband (SWA), Actiwatch and polysomnography (PSG). Our result indicates that HIIT has small-to-medium favorable effects on SE [SMD = 0.43, 95%CI (0.20, 0.65), *p* = 0.0002, *n* = 10] after removal of one study with high heterogeneity [34]. A total of 10 studies [20,21,22,25,26,28,30,36,37,39] compared SE pre- and post-intervention for a control group and experimental group included in the SE meta-analysis. Specifically, each individual study and aggregate ES for HIIT on SE are presented in Figure 6. Since I^2^ = 36%, fixed-effects model was used to calculate them. In addition, the adjusted funnel plot of SE is showed in Figure 5b, which appears to be asymmetrical. Both sensitivity and cumulative meta-analyses confirmed the reliability and stability of the findings. However, no significant effects were found in SL [SMD = 0.43, 95%CI (0.20, 0.65), *p* = 0.52, *n* = 7], TST [SMD = 0.08, 95%CI (−0.17, 0.33), *p* = 0.92, *n* = 9] and TB [SMD = −0.13, 95% CI (−0.48, 0.23), *p* = 0.48, *n* = 4].

### 3.5. Secondary Outcomes

Unfortunately, there were insufficient data (n < 5) to analyze the effects of HIIT on depression, anxiety, and health-related quality of life.

Table 2 shows the effects of HIIT on sleep of different outcomes.

### 3.6. Subgroup Analysis

As shown in Table 3, subgroup analyses were conducted for PSQI and SE. The results indicate that any type of HIIT can significantly reduce PSQI global scores [WMD = −0.53, 95%CI (−1.29, 0.24), *p* = 0.02, *n* = 3] [WMD = −1.62, 95%CI (−3.16, −0.08), *p* = 0.04, *n* = 6], while other type seems to be more effective; HIIT intervention duration >16 min and intervention lasts >8 weeks can significantly reduce PSQI global scores [WMD = −1.43, 95%CI (−3.09, 0.23), *p* < 0.0001, *n* = 5] [WMD = −1.10, 95%CI (−2.23, 0.03), *p* = 0.0009, *n* = 6]. Limited evidence indicates that HIIT frequency less than three times per week have better effects on PSQI global scores [WMD = −1.43, 95%CI (−3.09, 0.23), *p* < 0.0001, *n* = 3], compared with frequency high group. As for SE, subgroup analysis shows other types of HIIT was superior to cycling [SMD = 0.27, 95%CI (0.02, 0.52), *p* = 0.02, *n* = 7]. Low frequency HIIT intervention can bring more benefits [SMD = 0.53, 95%CI (0.26, 0.80), *p* = 0.007, *n* = 7], which is consistent with PSQI subgroup analysis [WMD = −2.13, 95%CI (−3.68, −0.59), *p* = 0.0001, *n* = 6]. There is no significant difference between different length subgroups for SE. Effects were significant for patients regardless of PSQI global scores and SE [WMD = −0.98, 95%CI (−1.90, −0.06), *p* = 0.04, *n* = 6] [SMD = 0.56, 95%CI (0.18, 0.95), *p* = 0.04, *n* = 3].

## 4. Discussion

The present study aimed to investigate the effects of HIIT on sleep. We observed that HIIT has large beneficial effects on sleep quality, reflected by PSQI global scores and small-to-medium favorable effects on SE. Furthermore, our subgroup results indicated that HIIT intervention sessions, regardless of type, that are performed longer than 16 min per session and continued throughout a period greater than eight weeks can significantly improve sleep quality, as indicated by lower PSQI global scores. This is particularly the case for patients. Our exciting findings help inform an optimal exercise prescription for improving sleep quality, which is in line with much of the extant literature. Lastly, to a certain extent, the sensitivity analysis mitigated the heterogeneity of included studies. Therefore, HIIT can be considered as an efficient and feasible way to achieve the benefits of sleep for adults.

For the primary outcomes, our main findings are consistent with most of the previous studies. Although some researchers believe that high-intensity exercise may elicit heightened physiological arousal and muscle soreness, counteracting the potential beneficial effects of exercise on sleep [44], meta-analysis results show the significant promoting effect of HIIT on overall sleep quality and sleep efficiency. More specifically, subgroup analysis results support some studies that have shown that one single bout of HIIT may increase psychophysiological stress and core body temperature, shift the circadian phase, or cause sympathetic hyperactivity [45]. However, if the training duration is too short, this interference effect on sleep will not appear. In addition, total sleep time (TST), some studies also defined as sleep duration, refers to the amount of time spent in bed asleep, while time in bed (TB) refers to the period between bedtime and wake time. TB seems to decline while TST remains unchanged, which means the reduction of wake time and further reflects the improvement of SE (TST expressed as a percentage of TB) from the side.

For the secondary outcomes, limited studies reported that there was no significant influence of HIIT on depression and anxiety. However, some authors have indicated that HIIT does have a positive effect on specific populations [46]. The reason why our present meta-analysis does not support this finding may be the heterogeneity of the population. Since research looking at the mechanism is lacking, we have a poor understanding of how emotion is influenced by HIIT. The influence may be explained in part by endocrine responses related to enjoyment feelings (e.g., serotonin secretion) and an increase of brain-derived neurotrophic factors (BDNF) [47]. It is worth noting that a number of studies have proved HIIT elicited similar or higher enjoyment and adherence levels than moderate-intensity continuous training [48,49]. In a word, HIIT should be concerned to help patients with mood barriers. A larger sample and high-quality randomized controlled trials are needed to verify HIIT’s advantages on depression and anxiety.

Although the various mechanisms have been discussed in many studies, how HIIT affects sleep is understudied. There is a need for studies designed to elucidate underlying mechanisms of functional adaptation to exercise, as well as define the best intensity, modality, and volume of HIIT. Moreover, future studies should focus on every stage of human sleep. For example, previous meta-analysis of the extant literature reached an agreement that acute exercise had a statistically significant but small increase in slow-wave sleep (SWS) and total sleep time (TST). However, the effect on SL and the amount of wake time after sleep onset (WASO) or rapid eye movement (REM) still need to be further explored [50,51].

The present research has several strengths. First, to the best of our knowledge, this is the only study to have used a quantitative approach to explore the effects of HIIT on sleep, and this approach provided reliable evidence that HIIT exerts a favorable effect on sleep. Second, our analyses included the most up-to-date literature. In fact, 57% of the included studies were published in 2020. Third, we included a number of RCT and crossover studies, regardless of the publication language, to summarize the causal relationship between HIIT and sleep quality. Lastly, the reliability of our findings is based on multiple forms of evidence since we included subgroup analyses, sensitivity analysis, and Trim and Fill analyses.

Despite these considerable strengths, the present analysis is not without limitations. On the one hand, as with all meta-analyses, we were limited to data that have been published. In order to account for this, we were very inclusive in our search strategy even though there were not enough available studies to allow us to examine the entire range of potential modifying effects. On the other hand, outcome assessment method and HIIT interventions varied across the included studies and the participants ranged from athletes to patients which result in inevitable heterogeneity. However, the sensitivity analysis conducted did not find any one study to marginally modify the overall effect. Notably, the results of some sensitivity analyses need to be interpreted with caution because the Trim and Fill method may lead to overcorrection [52].

To date, most studies on HIIT and sleep are conducted in adults and most HIIT interventions in improving sleep quality focused exclusively on cycling. Given the multiple benefits of HIIT for health, future studies should include HIIT among children and adolescents to explore its effects on their overall health and sleep quality. More forms of HIIT and HIIT compared with other types of exercise should be included in future studies as well. Additionally, future studies should investigate how exercise interacts with sleep disorders medication (e.g., whether incorporating HIIT into the daily life of a medicated person with sleep disorders could contribute to reducing the drug dose).

## 5. Conclusions

Taken together, HIIT can be a promising alternative for the treatment of sleep disorders. A large increase in sleep quality reflected by PSQI global scores and a small-to-medium favorable effect on SE were found after a HIIT intervention. A HIIT intervention duration of > 16 min and an intervention lasting >8 weeks can significantly reduce PSQI global scores. As for SE, subgroup analysis shows other types of HIIT was superior to cycling and HIIT intervention frequency ≤3 times per week can bring more benefits. Moreover, HIIT may help patients as well as the general population. There is a need for more rigorous and well-designed randomized controlled trials testing the effects of HIIT on sleep.

## Figures and Tables

**Figure 1 ijerph-18-10973-f001:**
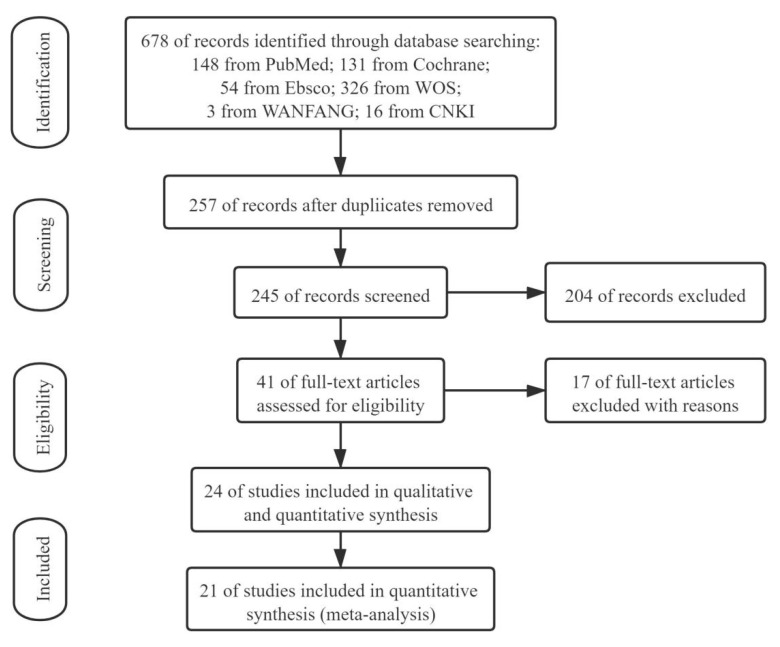
Flow diagram for selection of articles.

**Figure 2 ijerph-18-10973-f002:**
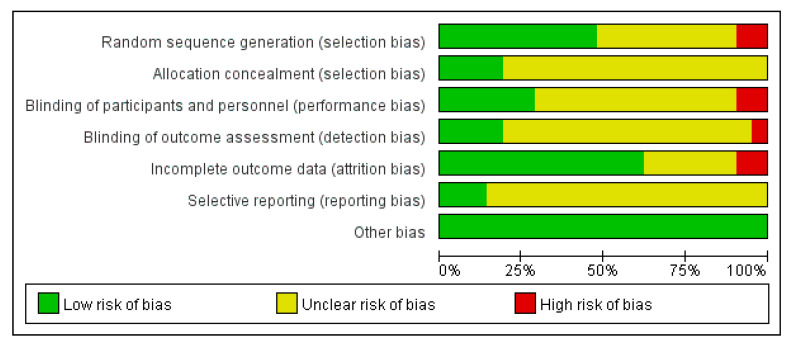
Risk of bias graph.

**Figure 3 ijerph-18-10973-f003:**
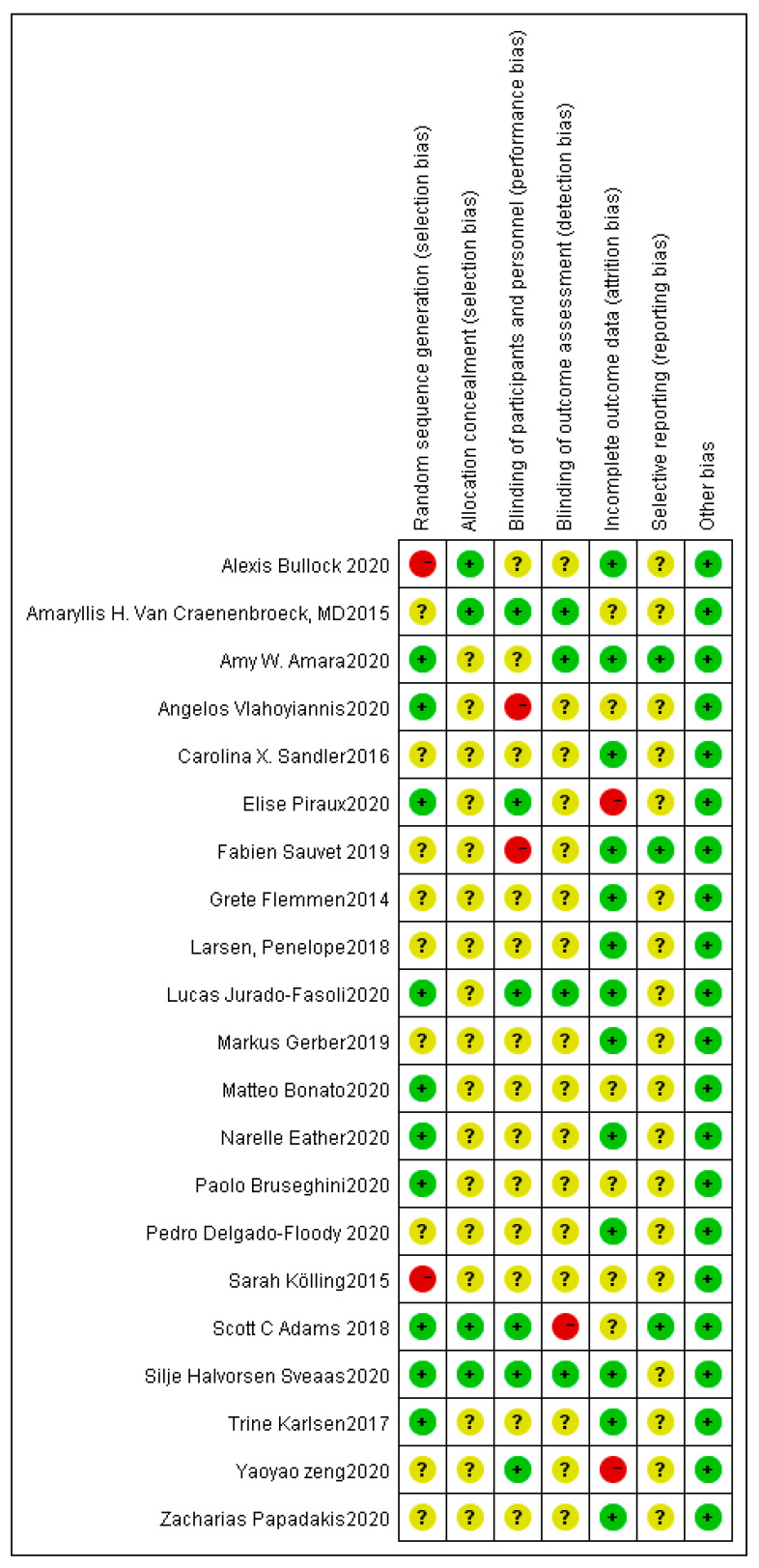
Risk of bias summary.

**Figure 4 ijerph-18-10973-f004:**
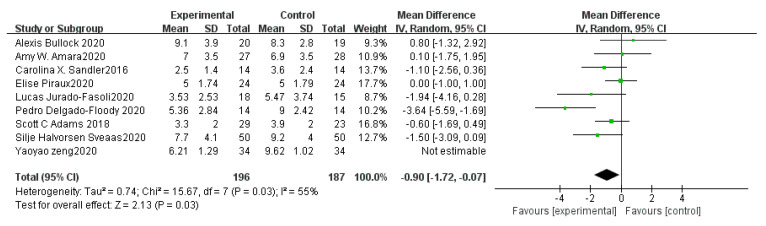
Forest plot of PSQI.

**Figure 5 ijerph-18-10973-f005:**
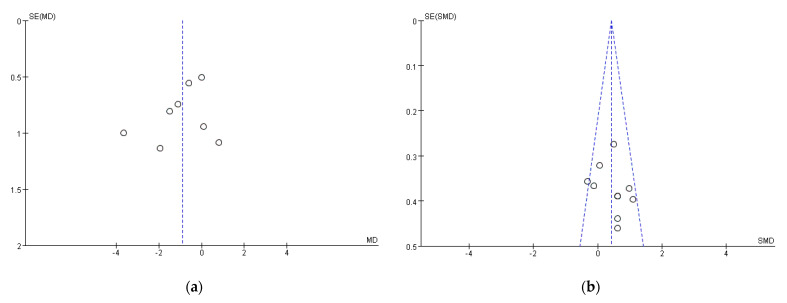
(**a**) Funnel plot of SE; (**b**) Funnel plot for SE.

**Figure 6 ijerph-18-10973-f006:**
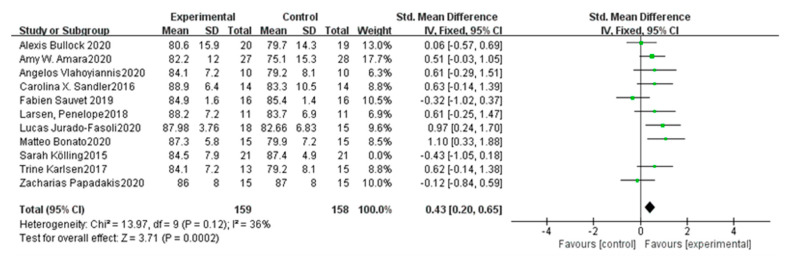
Forest plot of SE.

**Table 1 ijerph-18-10973-t001:** Characteristics of studies.

Author	Year	Sample Size (T/C)	Age of Participants (T/C)	Source of Participants (T/C)	HIIT Type	HIIT Intensity	HIIT Duration	HIIT Frequency	HIIT Length	Control Group	Primary Outcome	Secondary Outcome
Yaoyao Zeng [33]	2020	34/34	37.73/36.75	Rheumatoid arthritis	other	50% VO_2_max/75% VO_2_max	20 min/24 min/28 min	3 times per week	8 weeks	Health education	PSQI	SDS
Amaryllis H. Van Craenenbroeck, MD [32]	2015	19/21	51.5/54.7	Patients with CKD	cycling	90% of the heart rate achieved at the anaerobic threshold	10 min × 4	everyday	3 months	Usual care	KDQoL-SF	
Narelle Eather [19]	2020	24/23	43	Sedentary	Other	Weeks 1–4: 30:30, weeks 5–8: 40:20	8 min	2–3 times per week	8 weeks	Sedentary	Sleep patterns	BMI, motivation to exercise, etc.
Fabien Sauvet [36]	2019	16	27.3	Healthy young man TSD	Other	95–100% HRR	5 × 1 min	3 times per week	7 weeks	TSD	Actiwatch, PSG, sleep diary	
Scott C. Adams [29]	2018	29/23	44.0/43.3	Testicular cancer survivors	Other	5% of ventilatory threshold	35 min (4 × 4)	3 times per week	12 weeks	Usual care	PSQI	SF-36, fatigue, anxiety, depression
Elise Piraux [31]	2020	24/24	67.4/71.9	Prostate cancer patients	Cycling	≥85% THRmax	60 s × 8	3 times per week	5–8 weeks	Usual care	PSQI, ESS, ISI	FACT-G, etc.
Paolo Bruseghini [27]	2020	12/12	69.4/69.67	Healthy, active older adult men	Cycling	85–95% VO_2_max	2 min × 7	3 times per week	8 weeks	MICT	SWA	MET
Angelos Vlahoyiannis [39]	2020	10	23.9	Recreationally trained male	Cycling	90% Wmax	1 min × 10	Cross over once per week	4 weeks	CON/MICT/SIT	SWA	MET
Larsen, Penelope [20]	2018	11	49	Overweight, inactive men	Cycling	60 s work at 100% VO_2_peak: 240 s rest at 50% VO_2_peak	30 min	Cross over once per week		Self	PSG, diary	Appetite, etc.
Trine Karlsen [22]	2017	13/15	52.5/49.9	Obesity patients with moderate to severe obstructive sleep apnoea	Other	90–95% HRmax	4 min × 4	Twice per week	12 weeks	continue their normal lifestyle	ESS	
Alexis Bullock [30]	2020	20/19	72.4/72.3	Healthy sedentary elderly	Other	90–95% HR peak	4 min × 4	3 times per week	12 weeks	MICT	PSQI	
Pedro Delgado-Floody [38]	2020	14	40.7	Morbidly obese patients with poor sleep quality	Cycling	RT + HIIT	60 s × (4–7)	Twice per week	20 weeks	Self	PSQI	HRQoL, etc.
Lucas Jurado-Fasoli [40]	2020	18/15	53.1/51.7	Sedentary middle age	Cycling	>95% VO_2_max	40–65 min/wk	Twice per week	12 weeks	Maintain their lifestyle	PSQI, accelerometer	
Zacharias Papadakis [25]	2020	15	31	Healthy male	Other	3:2 intervals at 90% and 40% of VO_2_ reserve	500 kcals	Wash out at least 72 h	once	Self	SWA	
Sarah Kölling [34]	2015	21/21	22.8/23.6	Athlete	Other	6 × 4 s	30–15 Intermittent Fitness Test	Twice per day	6 days	Strength training	SWA, diary	
Markus Gerber [35]	2019	53	36.3	Patients with depression	Cycling	80% of maximal power output	30 s × 25	3 times per week	12 weeks	CAT	7-item Insomnia Severity Index	FEPS
Carolina X. Sandler [21]	2016	14	32	Unable to work, but not bed-bound	Cycling	75–80% APHRM	100 s × 5	Cross over	once	CONT	PSQI	
Matteo Bonato [28]	2020	15/15	25/23	Nonprofessional male football player	Other	90% to 95% HRpeak	4 × 4 min		once	SSGs	Actiwatch	
Amy W. Amara [26]	2020	27/28	65.33/65.82	Parkinson’s patients	Other	Body weight		3 times per week	16 weeks	Sleep hygiene intervention	PSG, PSQI, ESS	
Silje Halvorsen Sveaas [41]	2020	50/50	45.1/47/2	Patients with axial spinal arthritis	Cycling	90–95% of maximal heart rate	4 × 4 min	3 times per week	3 months	Do not intervene	PSQI	SF-36, GHQ12, EuroQol
Grete Flemmen [23]	2014	9/7	33/31	Patients with drug use disorders	Other	90–95% of HRmax	4 × 4 min	3 times per week	8 weeks	Usual care	ISI	

Abbreviations:T/C:HIIT group/Control group; PSQI: Pittsburgh Sleep Quality Index; PSG: polysomnography; SDS: Self-rating Depression Scale; CKD: Chronic Kidney Disease; KDQoL-SF: Kidney Disease Quality of Life-Short Form; TSD: total sleep deprivation; FEPS: Fragebogen zur Erfassung allgemeiner Persönlichkeitsmerkmale Schlafgestörter; SF-36: Short Form 36-item health survey; ESS: Epworth Sleepiness Score; ISI: Insomnia Severity Index; HRQoL: Health-related Quality of Life; AHI: Apnoea-hypopnea Index; SWA: Sense Wear Armband; MET: Metabolic Equivalents; FACT-G: Functional Assessment of Cancer Therapy–General; MICT: Moderate-intensity Continuous Training; SIT: Sprint Interval Training; CAT: continuous aerobic exercise training; CONT: Continues to be moderate; CON: non-exercise; SSGs: small-sided games.

**Table 2 ijerph-18-10973-t002:** Effects of HIIT on sleep of different outcomes.

	Chi^2^	df (*p*)	I^2^ (%)	Z (*p*)	N (E/C)	WMD (95%CI)	SMD (95%CI)
PSQI	15.67	7 (*p* = 0.03 *)	55	(*p* = 0.03 *)	196/187	(−1.72, −0.07)	-
ISI	1.37	2 (*p* = 0.50)	0	(*p* < 0.0001 **)	86/84	−1.65 (−2.43, −0.87)	-
SE	13.97	9 (*p* = 0.12)	36	(*p* = 0.0002 **)	169/158	-	(0.20, 0.65)
SL	11.59	6 (*p* = 0.07)	48	0.65 (0.52)	126/128	-	0.08 (−0.17, 0.33)
TST	12.14	8 (0.15)	34	0.10 (0.92)	160/159	-	0.01 (−0.21, 0.23)
TB	5.41	3 (*p* = 0.14)	45	0.70 (*p* = 0.48)	62/62	-	−0.13 (−0.48, 0.23)

I^2^: I squared statistic; Z: overall effect; N: Number; E: Experimental group; C: Control group; WMD: weighted mean difference; SMD: standardized mean difference; PSQI: Pittsburgh Sleep Quality Index; ISI: Insomnia Severity Index; SE: sleep efficiency; SL: sleep latency; TST: total sleep time; TB: time in bed. * *p* < 0.05; ** *p* < 0.01.

**Table 3 ijerph-18-10973-t003:** Effects of HIIT on sleep with different moderators.

	Subgroup	Chi^2^	df (*p*)	I^2^ (%)	Z (*p*)	N (E/C)	WMD (95%CI)	SMD (95%CI)
PSQI	Type (cycle)	2.52	2 (*p* = 0.28)	21	2.30 (*p* = 0.02 *)	103/97	−0.53 (−1.29, 0.24)	-
	Type (other)	31.55	5 (*p* < 0.00001 **)	84	2.08 (*p* = 0.04 *)	127/124	−1.62 (−3.16, −0.08)	-
PSQI	Duration (low)	10.72	2 (*p* = 0.005 **)	81	0.92 (*p* = 0.36)	52/52	−1.43 (−3.34, 0.49)	-
	Duration (high)	33.02	4 (*p* < 0.00001 **)	88	4.22 (*p* < 0.0001 **)	151/141	−1.43 (−3.09, 0.23)	-
PSQI	Frequency (low)	4.18	2 (0.12)	52	3.79 (*p* = 0.0001 **)	46/43	−2.13 (−3.68, −0.59)	-
	Frequency (high)	58.70	5 (*p* < 0.00001 **)	91	0.56 (*p* = 0.57)	184/178	−0.86 (−2.48, 0.77)	-
PSQI	Length (low)	38.01	2 (*p* < 0.00001 **)	95	0.94 (*p* = 0.35)	72/72	−1.54 (−3.94, 0.86)	-
	Length (high)	12.71	5 (*p* = 0.03 *)	61	3.33 (*p* = 0.0009 **)	158/149	−1.10 (−2.23, 0.03)	-
PSQI	General population	3.05	1 (*p* = 0.08)	67	0.40 (*p* = 0.69)	38/34	−0.55 (−3.23, 2.14)	
	Patients	12.52	5 (*p* = 0.03 *)	60	2.09 (*p* = 0.04 *)	158/153	−0.98 (−1.90, −0.06)	
SE	Type (cycle)	7.76	3 (*p* = 0.05)	61	1.35 (*p* = 0.18)	55/57	-	0.45 (0.07, 0.83)
	Type (other)	12.29	6 (*p* = 0.06)	51	2.06 (*p* = 0.04 *)	125/122	-	0.27 (0.02, 0.52)
SE	Duration (low)	14.14	6 (*p* = 0.03 *)	58	1.46 (*p* = 0.14)	111/110	-	0.13 (−0.14, 0.40)
	Duration (high)	1.14	3 (*p* = 0.77)	0	1.69 (*p* = 0.09)	57/56	-	0.84 (0.45, 1.23)
SE	Frequency (low)	4.74	6 (0.58)	0	2.70 (*p* = 0.007 **)	108/108	-	0.53 (0.26, 0.80)
	Frequency (high)	11.98	4 (*p* = 0.02 *)	67	1.03 (*p* = 0.30)	82/81	-	0.09 (−0.22, 0.40)
SE	Length (low)	14.08	5 (*p* = 0.02 *)	64	1.26 (*p* = 0.21)	91/91	-	0.14 (−0.16, 0.44)
	Length (high)	3.63	4 (*p* = 0.46)	0	1.91 (*p* = 0.06)	89/88	-	0.51 (0.21, 0.82)
SE	General population	9.01	5 (*p* = 0.11)	45	1.57 (*p* = 0.12)	90/86		0.24 (−0.06, 0.54)
	Patients	0.09	2 (*p* = 0.96)	10	2.91 (*p* = 0.004 *)	54/57		0.56 (0.18, 0.95)

I^2^: I squared statistic; Z: overall effect; N: Number; E: Experimental group; C: Control group; WMD: weighted mean difference; SMD: standardized mean difference; PSQI: Pittsburgh Sleep Quality Index; SE: sleep efficiency. * *p* < 0.05; ** *p* < 0.01.

## Data Availability

The data that support the findings of this study are available from the corresponding author, upon reasonable request.

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
