# Peer review of "Effects of High-Intensity Interval Training on Sleep: A Systematic Review and Meta-Analysis"

_ijerph, 2021, doi:10.3390/ijerph182010973_

Round 1

Reviewer 1 Report

Dear authors,
Your systematic review and meta-analysis have impressed me a lot. Really, poor sleep quality in modern society is a very big problem, and its relationships with lifestyle and physical activity are widely discussed. You clearly have shown high-intensity interval training (HIIT) as an effective alternative for health and sleep improvements for adult people, as well as a lack of knowledge about it. Your contribution to this field is very needed and valuable for health care.  At the same time, some comments should be corrected:
A)  Abstract recommended make structured, and include the following headings: Background (optional); Objective; Data sources; Study selection (or Review methods); Data extraction and Data synthesis; Results; Limitations (optional); and Conclusions.
B) In the section "Materials and Method" should include the following subheadings: Protocol and registration; Data collection process; Data items.
C) The subsection “Study selection” from the Section “Results” should move into the section “Materials and Methods”. 
D) The section “References” should be revising in accordance with the Instructions for Authors (see https://www.mdpi.com/journal/ijerph/instructions#front).
Thus, I’m pleased to say that I can recommend this important work to publish in the journal “International Journal of Environmental Research and Public Health” after revision and elimination of my comments. 

Author Response

Dear reviewer

Thank you for your recognition and criticism of our work. In this version,
A) Abstract have made structured, following PRISMA guidelines.
B) In the section "Materials and Method" , we included the following subheadings: Protocol and registration; Data collection process.
C) The subsection “Study selection” from the Section “Results” have moved into the section “Materials and Methods”. 
D) The section “References” have been revised in accordance with the Instructions for Authors (see https://www.mdpi.com/journal/ijerph/instructions#front).
Best wishes!

Reviewer 2 Report

  1. Please have this manuscript edited by a native speaker of the English language. At present, it contains a number of sentences with syntax errors. A few of the many instances are: 1) the difficult-to-understand sentence that reads “A new study has investigated the detrimental associations of poor sleep with all-cause and cause- specific mortality risks are exacerbated by low physical activity, suggesting likely synergistic effects…”; 2) the sentence that contains a number mismatch “Characteristics of the selected studies are presented in Table 1. Of them, 3 study were 125 conducted in Australia…”; 3) the sentence that contains a grammatical error “… random-effects model was used and funnel plot of PSQI was showed in figure 6…”; and 4) the sentence that contains a word (researches) that is somewhat atypical in English writing “Although the various mechanisms have been discussed in many researches…”.
  2. Please provide an explanation for exactly why the number of acceptable articles fell from 257 to 41 as indicated in the first paragraph of the Results section.
  3. Please clarify the statement that “Lastly, there was one study conducted follow-up diagnoses.” Follow-up diagnoses as opposed to what?
  4. Please translate the non-English characters present in Table 1 and Figure 3. Also, in Table 1, please correct the capitalization in the participant-description column and indicate that the participants in the Sandler study were “Patients” unable to work…
  5. Under section 3.4, please indicate that lower scores on both the subscales of the PSQI and the total score on the PSQI are indicative of better sleep (and less medication usage).
  6. Please describe the Insomnia Severity Index in the Primary Outcomes section since you appropriately described the PSQI in this section.
  7. Perhaps it is just my ignorance of the meta-analysis process that is creating some confusion, but I don’t understand the sentence that reads:” Moreover, there were not enough available studies in the category of Insomnia Severity Index 169 (ISI) [WMD=-1.65, 95%CI (-2.43, -0.87), p<0.0001, n=3], although the effects were highly statistically significant.” This statement seems to contain contrary information (i.e., “not enough data to evaluate” versus “effects were highly significant”). Please clarify.
  8. Please clarify what is meant by “other type” in the sentence which appears in the Subgroup Analysis section (i.e., the sentence that says “…while other type seems to be more effective; HIIT intervention duration > 16 minutes and intervention lasts > 8 weeks can significantly reduce PSQI global scores”).
  9. The first sentence in the Discussion section should indicate the size of favorable effects associated with PSQI scores as well as the size of favorable effects associated with objective Sleep Efficiency scores.
  10. Please consider rewording the second and third sentences in the Discussion section to read: “Furthermore, our subgroup results indicated that HIIT intervention sessions, regardless of type, that are performed longer than 16 minutes per session and continued throughout a period greater than 8 weeks can significantly improve sleep quality as indicated by lower PSQI global scores. This is particularly the case for patients.”
  11. Please clarify (or elaborate upon) the fourth sentence in the Discussion section which reads “Lastly, to a certain extent, the sensitivity analysis mitigated the heterogeneity of included studies.” At present, the exact point being made is unclear.
  12. Consider rewording the second sentence in the third paragraph of the Discussion section to read “However, some authors have indicated that HIIT does have a positive effect on specific populations [38].” Also, clarify whether or not the present meta-analysis supports this finding.
  13. I recommend modifying the paragraph on the study’s strengths to read: “The present research has several strengths. First, to the best of our knowledge, this is the only study to have used a quantitative approach to explore the effects of HIIT on sleep, and this approach provided reliable evidence that HIIT exerts a favorable effect on sleep. Second, our analyses included the most up-to-date literature. In fact, 57% of the included studies were published in 2020. Third, we included a number of RCT and crossover studies, regardless of the publication language, to summarize the causal relationship between HIIT and sleep quality. Lastly, the reliability of our findings is based on multiple forms of evidence since we included subgroup analyses, sensitivity analysis, and Trim and Fill analyses.” I would then join this paragraph to the limitations paragraph, starting with “Despite these considerable strengths, the present analysis is not without limitations. On the one hand, as with all meta-analysis, we were limited to data that have been published. In order to account for this, we were very inclusive in our search strategy even though there were not enough available studies to allow us to examine the entire range of potential modifying effects... ”
  14. A general issue the authors should consider is whether they wish to maintain the focus on improved sleep quality throughout the article rather than transitioning to the treatment of sleep disorders towards the end of the manuscript. The first mention of “sleep disorders” occurs on line 261 of the next-to-the-last page of the text, and I found this to be somewhat inconsistent with the more general focus on “sleep quality” that was present in the majority of the preceding material.

Author Response

Dear reviewer

Thank you for your recognition and criticism of our work. In this version,

  1. We have corrected many grammar problems.
  2. Why the number of acceptable articles fell from 257 to 41 as indicated in the first paragraph of the Results sectionhave shown in line 138-139.
  3. Sincethe study conducted follow-up diagnoses is so limited (only one), we didn’t report it.
  4. We havetranslated the non-English characters present in Table 1 and Figure 3. We also have corrected the capitalization in the participant-description column and indicate that the participants in the Sandler study were “Patients” unable to work…
  5. Under section 3.4, We cited a classic literatureto indicate that lower scores on both the subscales of the PSQI and the total score on the PSQI are indicative of better sleep (and less medication usage).
  6. We added explanation of Insomnia Severity Index in the Primary Outcomes section.
  7. We have reworded the sentenceseems to contain contrary information (i.e., “not enough data to evaluate” versus “effects were highly significant”).
  8. Please clarify what is meant by “other type” in the sentence which appears in the Subgroup Analysis section (i.e., the sentence that says “…while other type seems to be more effective; HIIT intervention duration > 16 minutes and intervention lasts > 8 weeks can significantly reduce PSQI global scores”).
  9. The first sentence in the Discussion section reasserted the aim of this study. The second sentence haveindicated the size of favorable effects associated with PSQI scores as well as the size of favorable effects associated with objective Sleep Efficiency scores.
  10. In the Discussion section, we have reworded the second and third sentences, and the second sentence in the third paragraphfollowed your valuable suggestions
  11. To a certain extent, the sensitivity analysis mitigated the heterogeneity of included studiesby Trim and Fill analyses.
  12. We havemodified the paragraph on the study’s strengths and limitations using the expressions you recommended.
  13. Wedo wish to maintain the focus on improved sleep quality throughout the article, so we have revised some inappropriate statements about sleep disorders. Finally, the first mention of “sleep disorders” occurs on line 34 in order to emphasize the importance of this article. The second mention of “sleep disorders” in the text occurs in the last paragraph of discussion in order to expand its application value.

Best wishes!

Reviewer 3 Report

GENERAL COMMENTS:

This is an interesting review of studies that have assessed the effects of high-intensity interval training (HIIT) on sleep of different populations. My major comment is for the authors to fully follow the PRISMA guidelines to adequately and systematically report the review work. The specific comments have been suggested under the various sections of the manuscript:

ABSTRACT/SUMMARY:

  • It is important for authors to follow the PRISMA 2009 guidelines for all sections of the manuscript, including the abstract.
  • Authors should specify the population.
  • How was risk of bias assessed and what tool did the authors used? This is very important and should be stated in the abstracts as well.
  • Was the protocol for the systematic review and meta-analysis registered? If so, the details of the registration should be provided.
  • Keywords: “HIIT” should be written in full.
  • Conclusion: the second sentence of the conclusion is not clear and should be re-written. What was the population(s) of the included studies?

INTRODUCTION:

  • Page 1, lines 33-35: “sleep time” should be “sleep duration”; “standard” should be “recommended sleep duration”. Properly write “1/3”. Please rephrase these lines.
  • The physiological, psychological and physical effects of HIIT should be described in the introduction.
  • Authors must properly explain why HIIT may be a better option compared to moderate-intensity exercise.
  • The following papers may be helpful:
  • The effects of evening high-intensity exercise on sleep in healthy adults: A systematic review and meta-analysis:https://doi.org/10.1016/j.smrv.2021.101535 and DOI: 1123/ijspp.2019-0498

METHODS:

  • Please duly follow the PRISMA guidelines. Authors must use the PRISMA 2009 checklist.
  • Including criteria: were healthy non-athletes included in the review? The populations considered should be fully described. Give the specific age range considered.
  • Sleep outcome measures should be fully stated: “(Pittsburgh Sleep Quality Index, etc.)” is unacceptable, please fully state them. List all the objective sleep assessments tools used.
  • Assessment of quality: authors should state the specific risk of bias assessment tool used.
  • Page 3: Statistical analysis: paragraph 1: authors should completely list all items; for example; line 93: study characteristics “(source, first author’s name, year of publication, etc.)” should be fully stated. This important for reproducibility.

RESULTS:

  • The qualitative results should be described separately from the quantitative. Results should be re-written and re-structured.
  • Please follow the PRISMA guidelines for the results section too.
  • Lines 118-119 should be rephrased.
  • Figure 1: Identification – numbers from the various databases should be separated by semi-colon.
  • Study characteristics: lines 125-130 as well as lines 133-135, provide the corresponding references again each country.
  • Table 1: include height, weight and BMI of the participants. Under HIIT type, “other”; please state the specific exercise type.
  • Primary outcomes: all terms should have already been explained by now. Please follow the PRISMA guidelines. Provide the references for the 8 articles.
  • Figure 4: the Chinese word should be translated in English. Line 177-178; provide the reference for the study with high heterogeneity.
  • Table 2: all abbreviations should be fully defined under the table. Why were SL, TST and TIB analyzed with SMD?
  • Subgroup analysis: paragraph 1 should not be here. This should have already been explained in the methods of synthesis. Please follow the PRISMA 2009 guideline and provide every detail.
  • What sensitivity analyses were conducted? Fully explain in the methods section and also show results at the results sections. Please follow the PRISMA guidelines as strongly recommended.
  • Figure 6: Funnel plot; are both plots made from the same data? if so maintain one, for example plot B.

DISCUSSION:

  • Good; but authors must fully discuss the results of the subgroup analysis.

REFERENCE:

  • Reference 43 should be re-written

Author Response

Dear reviewer

Thank you for your recognition and criticism of our work. In this version, we followed the PRISMA 2020 guidelines for all sections of the manuscript and used PRISMA checklist.

 ABSTRACT

  • We specifiedthe population (adults).
  • We clarified how was risk of bias assessed and what tool did weused.
  • Theprotocol for the systematic review and meta-analysis was registered and the details of the registration were provided.
  • Keywords: “HIIT” waswritten in full.

INTRODUCTION:

  • Page 1, lines 33-35: “sleep time” was replaced by“sleep duration”; “standard” was replaced by “recommended sleep duration”.
  • The physiological, psychological and physical effects of HIIT weredescribed in the introduction.
  • Why HIIT may be a better option compared to moderate-intensity exercisewas explain
  • The papers you recommended were muchhelpful and we cited one of them in the introduction. 

METHODS:

  • Including criteria: healthy non-athletes were included in the review.The specific age of populations ranged from 18 to 75 years old.
  • Sleep outcome measures and all the objective sleep assessments tools werefully stated.
  • Assessment of quality: the specific risk of bias assessment tool usedwas provided by the Cochrane Handbook for Systematic Reviews of Interventions.
  • Statistical analysis: study characteristics and other items werefully stated.

RESULTS:

  • Results werere-structured and some sentences were
  • Figure 1: Identification – numbers from the various databases wereseparated by semi-colon.
  • Study characteristicsand Primary outcomes: provided the corresponding references.
  • Table 1: We didn’t add new column because some studies didn’t report height, weight and BMI of the participants. Under HIIT type, “other”used as a summary if the specific exercise type was not clearly pointed out to be cycling.
  • Figure 4: the Chinese word wastranslated in English. The study with high heterogeneity was provided corresponding reference.
  • Table 2: all abbreviations were fully defined under the table. SL, TST and TIB were analyzed with SMDbecause the data were obtained from different measurement tool
  • Subgroup analysis: paragraph 1 have been movedinto the methods of synthesis.
  • Sensitivity analyses were conductedusing different effect models, funnel plots,Trim and Fill analyses.
  • Figure 6: two funnel plots were made from differentdata.

DISCUSSION:

  • The results of the subgroup analysiswere fully discuss

REFERENCE:

  • Reference 43 should be re-written

Best wishes!

Round 2

Reviewer 3 Report

This shows a much improved version of the manuscript, but the PRISMA format was not fully followed. A few other minor corrections need to be done:

1) Introduction: line 36: please re-write this: "1/3 people".

2) Methods: statistical analysis: lines 122-123: please give a reference.

3) Titles and/or explanations for figures should be properly and fully described. Eg., fig 4 & 5.

4) Page 4: 220-225; are they cancelled? The lines are not legible.

5) Discussion: lines 264-265; "Total Sleep Time" and "Sleep Duration" should be re-written as "total sleep time" and "sleep duration" respectively.

I strongly recommend the authors fully follow the PRISMA guidelines for reporting their.

Author Response

Dear reviewer

Thank you for your recognition and criticism of our work. In this version, we fully followed the PRISMA guidelines for reporting

1) Introduction: line 36: "1/3 people" has been corrected.

2) Methods: statistical analysis: lines 122-123: has been given a reference.

3) Titles and/or explanations for figures have been properly and fully described. 

4) Page 4: 220-225; they are not canceled, hoping this manuscript is legible.

5) Discussion: lines 264-265; "Total Sleep Time" and "Sleep Duration" have been re-written as "total sleep time" and "sleep duration" respectively.
